# Chronic Neuroinflammation and Cognitive Decline in Patients with Cardiac Disease: Evidence, Relevance, and Therapeutic Implications

**DOI:** 10.3390/life13020329

**Published:** 2023-01-24

**Authors:** Jan Traub, Anna Frey, Stefan Störk

**Affiliations:** 1Department of Internal Medicine I, University Hospital Würzburg, 97080 Würzburg, Germany; 2Department of Clinical Research & Epidemiology, Comprehensive Heart Failure Center, University and University Hospital Würzburg, 97078 Würzburg, Germany

**Keywords:** neuroinflammation, cognitive impairment, dementia, myocardial infarction, heart failure, hypertension, coronary artery disease, atrial fibrillation, cardiac arrest, aortic valve stenosis

## Abstract

Acute and chronic cardiac disorders predispose to alterations in cognitive performance, ranging from mild cognitive impairment to overt dementia. Although this association is well-established, the factors inducing and accelerating cognitive decline beyond ageing and the intricate causal pathways and multilateral interdependencies involved remain poorly understood. Dysregulated and persistent inflammatory processes have been implicated as potentially causal mediators of the adverse consequences on brain function in patients with cardiac disease. Recent advances in positron emission tomography disclosed an enhanced level of neuroinflammation of cortical and subcortical brain regions as an important correlate of altered cognition in these patients. In preclinical and clinical investigations, the thereby involved domains and cell types of the brain are gradually better characterized. Microglia, resident myeloid cells of the central nervous system, appear to be of particular importance, as they are extremely sensitive to even subtle pathological alterations affecting their complex interplay with neighboring astrocytes, oligodendrocytes, infiltrating myeloid cells, and lymphocytes. Here, we review the current evidence linking cognitive impairment and chronic neuroinflammation in patients with various selected cardiac disorders including the aspect of chronic neuroinflammation as a potentially druggable target.

## 1. Introduction

Acute myocardial infarction (MI) is the leading cause of morbidity and mortality worldwide. It affects about 16 million patients every year [1] and predisposes not only to chronic heart failure (HF) [2], but also to an accelerated cognitive decline beyond ageing. Not only MI and HF, but also other common cardiac disorders such as hypertension, atrial fibrillation (AF) and coronary artery disease (CAD) markedly increase the risk for developing mild cognitive impairment (MCI), vascular cognitive impairment and dementia (VCID), and Alzheimer’s dementia (AD) [3,4]. Due to the ageing population and an increase in the number of patients with cardiac diseases, these sequelae are of great relevance as they impact both morbidity and quality of life on the side of patients and their relatives, and healthcare costs on the side of society [5]. However, the pathophysiological links between cardiac disease, central nervous system (CNS) dysfunction and cognitive decline remain poorly understood. Shared risk factors such as hypertension, diabetes, obesity, arteriosclerosis, age, and genetic predisposition [6] might be involved, but there are also specific cardiac sequelae such as a decreased cerebral blood flow, neurohumoral activation, and systemic inflammation [7]. There is now compelling preclinical and clinical evidence that the latter might also induce or deteriorate chronic inflammatory responses of the central nervous tissue, which mediate pathophysiological processes leading to neurodegeneration [8]. Thus, so-called ‘neuroinflammation’ might be partially responsible for cognitive decline after MI, but also in other cardiac disorders such as chronic HF [9]. The current review aims to summarize current knowledge on cognitive decline in cardiac disorders, introduce the concept of neuroinflammation, highlight its role and clinical relevance in cardiac disease, and discuss potential therapeutic options. 

## 2. Cognitive Decline and Dementia in Common Cardiac Disorders

### 2.1. Myocardial Infarction and Coronary Artery Disease

There is now consolidated evidence for an accelerated cognitive decline after MI [3,4]. The Rotterdam Study reported on 4971 individuals in whom a prior history of MI was associated with worse cognitive performance, independent of age and education [10]. Further analysis revealed an increased risk of dementia and a higher degree of cerebral small vessel disease in people with unrecognized MI [11]. Likewise, the Bronx Aging Study studying 488 subjects observed a five-fold increased risk of developing dementia when patients with a history of MI were compared to those without [12]. Consistently, an accelerated cognitive decline was observed only after and not prior to the occurrence of MI in the English Longitudinal Study of Ageing with 7888 patients [13]. Recurrent non-ST elevation MI was independently associated with cognitive decline after one year of follow-up in the “Improve cardiovascular outcomes in high-risk older patients” (ICON-1) study in 211 patients [14]. A history of MI doubled the risk for MCI or probable dementia in the Women’s Health Initiative Memory study in 6455 cognitively intact, postmenopausal women [15]. Likewise, in the Italian Longitudinal Study of Aging, a history of MI was associated with an increased risk of progression from MCI to dementia among 2963 participants [16]. Interestingly, in a huge Danish nationwide population-based cohort study with 314,911 patients with MI and 1,573,193 matched comparison cohort members, MI predicted a greater risk for VCID, but not for AD [17].

Cognitive decline beyond ageing also affects people with CAD. In a small Californian study including 74 subjects, CAD was associated with a greater decline in global cognition, verbal memory, and executive function [18]. Zheng et al. found that the presence of atheromatosis in coronary vessels positively correlated with cognitive dysfunction [19]. The rate of memory decline among older adults with CAD seems to be similar between groups of patients undergoing coronary revascularization with coronary artery bypass grafting or percutaneous coronary intervention [20]. Of note, some data also point towards the reversibility of the process of cognitive decline, e.g., 43 patients with CAD who were treated with a statin and weekly in-hospital aerobic exercise for 5 months improved their Mini-Mental State Examination (MMSE) score [21]. In addition, lower cardiovascular health predicted late-life decline in cognitive function among high-risk CAD patients [22]. 

### 2.2. Heart Failure

Chronic HF is a complex clinical syndrome that often coexists with multiple comorbidities. MCI appears to be a particularly important condition in HF [7], although reports on its prevalence vary between 25% and 75% across population-based studies due to variation in definitions and diagnostic criteria [23,24,25,26,27]. Patients with HF typically exhibit MCI in the domains of memory, working memory, attention, processing speed, and executive function [25,28]. In patients with HF, MCI was shown to significantly increase the risk of hospitalization and mortality, and decrease quality of life [7]. Chronic HF is also associated with an increased risk of dementia and AD in older adults [29]. On the contrary, a Danish nationwide population-based cohort study found that chronic HF was associated with an increased risk of all-cause dementia, but not AD [30]. Thus, there is no consensus as to whether HF may also increase the risk of AD. Impaired cognition in people with chronic HF was reported to relate to hypertension, daytime sleepiness, stress, and poor quality of life [31]. In this regard, type 2 diabetes was shown to accelerate MCI and significantly reduce the executive ability of elderly patients with chronic HF [32]. In contrast to chronic HF, MCI in acute HF remains frequently unrecognized, since there are still many unresolved issues regarding cognitive changes in patients hospitalized with acute HF [33].

### 2.3. Hypertension

Hypertension has been shown to be a risk factor for both AD and VCID, as indicated by several epidemiologic studies [34]. Cerebral infarcts, lacunae, and white matter changes are implicated in the pathogenesis of VCID, but may also favor the development of AD [35]. A recent investigation including 90 individuals found that hypertensive participants revealed more deficits in skills involving delayed recall and prefrontal-region skills [36]. Likewise, the greatest impact on cognitive function in those with hypertension appears to be on executive or frontal lobe function, similar to the area most damaged in vascular dementia [37]. In a large Chinese study with 2413 individuals, it was reported that hypertension diagnosed during mid-life was associated with worse cognition compared to that diagnosed in late life [38]. Managing and controlling blood pressure could thus preserve cognitive function, e.g., by reducing the risk of VCID or stroke [39].

### 2.4. Atrial Fibrillation

According to a recent meta-analysis including 2,415,356 individuals, AF is linked to an increased risk of incident dementia and cognitive decline [40] (random-effect hazard ratio 1.36). Further, prospective observational studies have shown that AF increased the risk of stroke, an important cause of cognitive impairment, although the association between both conditions may well be independent of stroke and other shared risk factors [41]. Along these lines, incidental AF was associated with an increased risk of both early and late-onset dementia, independent of the occurrence of clinical stroke [42]. Interestingly, the use of oral anticoagulants and successful catheter ablation were associated with a decreased risk of developing dementia [43]. In a post hoc sub analysis of the Systolic Blood Pressure Intervention (SPRINT) trial, processing speed was the most prominent cognitive domain affected by AF. Potentially, this property might help with screening for early signs of cognitive dysfunction [44]. Several markers indicative of atrial cardiomyopathy, a structural and functional disorder of the left atrium, such as increased brain natriuretic peptide and left atrial enlargement, were also associated with an increased risk for cognitive impairment [45].

### 2.5. Aortic Valve Stenosis

Comprehensive neurocognitive assessment unmasked advanced cognitive impairment in patients with severe aortic stenosis planned for transcatheter aortic valve replacement (TAVR) [46]. Interestingly, the mere presence of aortic valve calcification in computer tomography was not associated with cognitive impairment in any cognitive test, nor any measure of global cognition [47]. There is now also evidence that TAVR may improve cognitive functions that depend on cerebral perfusion, especially of the hippocampus in elderly patients with severe aortic stenosis [48]. Such preservation or improvement of cognition after TAVR is particularly encouraging, as this population is characterized by a rapidly declining cognitive trajectory set in motion by ageing [49].

### 2.6. Cardiac Arrest

After cardiac arrest, the initial survival of patients is limited by brain death and severe neurological damage, either mandatorily (in the case of brain death) or potentially (in the case of severe damage and corresponding presumed patient wish) [50]. Cognitive dysfunction, in particular memory problems, is frequent amongst survivors of out-of-hospital cardiac arrest [51], e.g., cognitive impairment four years after cardiac arrest was present in more than one-quarter of patients [52]. Cohort studies demonstrated a high prevalence (54.4%) of long-term cognitive deficits and functional limitations in cardiac arrest survivors [53], even in those with apparently favorable neurological recovery [54]. While early systematic testing of cognitive performance is recommended by the current post-resuscitation guidelines, such concepts are infrequently implemented [55]. 

## 3. Acute vs. Chronic Neuroinflammation

The CNS is considered an immunologically ‘privileged’ organ because peripheral leucocytes are blocked from entering by the blood–brain barrier. The term neuroinflammation refers to both acute and chronic inflammatory responses of the central nervous tissue. Neuroinflammation is central to the shared pathology of several acute and chronic brain diseases [56]. Acute neuroinflammation is usually caused by infection, trauma, stroke, or toxins, and characterized by platelet deposition, edema, and endothelial cell activation [57]. It includes the activation of resident microglia, resulting in a phagocytic phenotype and the release of cytokines and chemokines [58]. Acute neuroinflammation is typically short-lived and unlikely to be detrimental to long-term neuronal survival. Currently, it is believed that an acute neuroinflammatory response is beneficial to the CNS, since it tends to minimize further injury and contributes to the repair of damaged tissue [59]. 

In stark contrast, chronic neuroinflammation must be considered a long-standing and often self-perpetuating process after an initial injury or insult. It is characterized by persistent activation of microglia and results in sustained oxidative and nitrosative stress by resident glial cells [60]. It is often accompanied by a disrupted blood–brain barrier, which facilitates the infiltration of peripheral macrophages into the brain parenchyma, thus further perpetuating inflammation [61]. Chronic neuroinflammation may be caused by external (toxic metabolites, microbes, viruses, air pollution) or internal factors (ageing, autoimmunity), and is associated with neurodegenerative disorders [62]. Other than acute neuroinflammation, chronic neuroinflammation is typically detrimental and damaging to nervous tissues [59]. As the most abundant cell type within the CNS, resident microglia, which are part of the innate immune system, are responsible for immune scavenging, phagocytosis, antigen presentation, cytotoxicity, synaptic stripping, and the promotion of cell repair [63]. Therefore, microglia are currently regarded as the crucial point of convergence for multiple triggers eliciting an adaptive immune response [59]. Microglia undergo morphological, proliferative, and functional changes in response to the above-mentioned factors [9]. They can modify microglial cell-surface receptor expression, transforming the specific cell task from a monitoring role to one of protection and repair [58]. Secondarily, astrocytes may become activated in response to signals produced by activated microglia, release various growth factors, and undergo morphological changes themselves [57]. In conclusion, whether neuroinflammation affects the brain in a beneficial or harmful way may depend critically on the duration of the inflammatory response and the immune cells involved [59].

## 4. Chronic Neuroinflammation in Neurodegenerative Disorders

Common neurodegenerative disorders such as AD, but also multiple sclerosis, Parkinson’s disease, Huntington’s disease, amyotrophic lateral sclerosis, and tauopathies, are associated with chronic neuroinflammation and elevated levels of cytokines [59,64]. In these conditions, it is thought that microglia (upon other) produce factors such as interleukin-1, tumor necrosis factor alpha and nitric peroxide that are toxic to neurons [64]. Neuropathological and neuroradiological studies indicate that neuroinflammatory responses precede the significant loss of neuronal populations [65], rendering neuroinflammation a potential therapeutic target.

In AD brains, microglia display an activated phenotype when surrounding amyloid plaques [66] and produce cytokines and other pro-inflammatory mediators [67]. Although it is clear that not all microglia activation is injurious to neurons, it is becoming widely accepted that a certain phenotype of neurotoxic microglia—e.g., when overexpressing mutant superoxide dismutase [68]—has a central role in the pathophysiology of AD [59]. Furthermore, the loss of microglial neuroprotective and phagocytic functions, as indicated by the dysfunctional clearance of amyloid plaques, and correlations between microglial activation and tau tangle spread demonstrate the critical involvement of malfunctioning microglia in driving tau propagation [69]. Recent data from brain imaging support the association of increased neuroinflammation during the progression of MCI and AD [70]. In contrast to ‘classical’ AD, VCID is a heterogeneous brain disease, where chronic cerebral hypoperfusion and persistence of an ischemic and hypoxic micro-environment leads to neuroinflammation, oxidative stress, neurotrophic uncoupling, destruction of the blood–brain barrier, and eventually to cognitive dysfunction [71]. In several animal models of VCID, neuroinflammation caused by chronic cerebral hypoperfusion induced white matter damage, while microglia and astrocytes were activated [71,72].

Summing up, microglia are thought to act as critical sensors of a disturbed brain tissue homeostasis in neurodegenerative disease: their differential activation constitutes the pivotal point regulating neuroinflammation and inducing neurotoxicity and subsequent cognitive decline [73]. The emergence of novel techniques such as single-cell RNA sequencing revealed the heterogeneity of microglia, including the differentiation of microglia and CNS-associated macrophages [74].

Thus, phenotypical and functional changes to microglia and associated cells in cardiac disease may also precede or accelerate neurodegenerative processes, which may lead to higher rates of AD and VCID and accelerated cognitive decline in these patients. In the following, we summarize current evidence on neuroinflammatory modulations in cardiac disease.

## 5. Neuroinflammatory Sequelae of Cardiac Disorders

### 5.1. Myocardial Infarction and Coronary Artery Disease

Recently published positron emission tomography data were the first to image microglial activation early and late after MI in humans [75,76]. Here, increased signals of the mitochondrial translocator protein (TSPO), which is expressed on activated microglia, were seen in the cerebellum, temporal and frontobasal cortex, as well as the hypothalamus. The strategy to investigate neuroinflammation after MI is rooted in promising preclinical findings, which are summarized in the following.

Small animal studies used immunohistochemistry to detect regional phenotypical changes to microglia after MI: morphologically altered microglia after MI were observed in the rostral ventrolateral medulla (RVLM), the nucleus tractus solitarius (NTS), and the periaqueductal grey (PAG) in rats, which share important cardiovascular regulatory functions [77]. Concordantly, an increase in such activated microglia occurred in the hypothalamic paraventricular nucleus (PVN) 2 to 16 weeks after MI, but not earlier [78,79]. These studies suggest that the observed changes are correlates of an augmented sympathetic tone after MI.

However, murine (and human) TSPO imaging revealed elevated neuroinflammatory signals even beyond the aforementioned regions [75]. Along these lines, 48 h after MI, significant changes in specific microglia phenotypes in the PVN, thalamus, prefrontal cortex, and hippocampus were found through the immunofluorescence approach [80]. Two studies demonstrated increased microglial activity peaking at day 3 in the caudate putamen and hippocampus after 30 min ischemia/reperfusion injury [81,82]. Further supporting the idea of global chronic inflammation in the CNS, pro-inflammatory cytokine levels in the brain were elevated following MI [83], and an augmented cerebral TNF-α expression of microglia was seen in a murine study 6–8 weeks after cardiac ischemia.

Activation of microglia also triggers the subsequent activation of neighboring astrocytes [84]. Positron emission tomography with ^11^C-methionine identified astroglial activation after permanent coronary artery ligation in mice [76]. These findings were paralleled by experiments, where astrocytes were activated in PVN with enhanced expression of cytokines and glial fibrillary acidic protein (GFAP) [85]. Five weeks after experimental MI, there was a sharp increase in GFAP-positive astrocytes in the hippocampus in rats [86].

Lastly, infiltrating monocyte and neutrophil abundance was also increased in the brain within the first day after MI and remained elevated for at least one week compared to controls [87]. As a possible mechanism, the skull bone marrow was found to contain direct vascular channels to the brain parenchyma, facilitating the migration of monocytes and neutrophils, which are transcriptionally distinct from host microglia [88], into the brain parenchyma after CNS injury [89].

Lymphocytes are also likely involved in neuroinflammatory changes after MI. Recent studies suggested that CD4^+^ T lymphocytes exhibit biphasic kinetics post MI [90] with a rapid CD4^+^ T cell response at 3 days post-MI and a second phase of activation in chronic HF. Increased levels of pro-inflammatory markers after MI were shown to be associated with worse systolic function [91]. It has also been established that peripherally activated T cells are capable of adhering to and crossing the blood–brain barrier to enter the perivascular space [92]. Murine and human immunohistochemical analysis showed a crucial role of perivascular and parenchymal infiltrating CD4^+^ helper T cells, but not cytotoxic T cells (CD8^+^) or B cells (CD20^+^), in the neocortex, hippocampus, and striatum in neurodegenerative disease [93], with close proximity to activated astroglia, microgliosis, and expression of pro-inflammatory cytokines. Therefore, peripherally activated T cells may also trigger or even amplify neuroinflammation after MI.

### 5.2. Heart Failure

The above-mentioned long-lasting elevated TPSO signals after MI [75] suggest relevant global chronic neuroinflammation also in chronic (ischemic) HF. Animal models of HF demonstrated that the expression of inflammatory genes, such as Toll-like receptor-4 (TLR-4), tumor necrosis factor-α, and interleukin-6, were significantly upregulated in the cortex and hippocampus, particularly in mice [94]. Most preclinical studies in HF to date, however, focused on autonomic control areas of the CNS, as there is some evidence suggesting that it plays an important role in cardiac dysfunction in HF [95].

There are three primary brain nuclei involved in the regulation of sympathetic tone, i.e., NTS, PVN, and RVLM [96]. In fact, the chronic activation of RVLM astrocytes is associated with increased mortality in animals exhibiting HF with a reduced left ventricular ejection fraction [97]. Indeed, activated microglia have been observed in these key autonomic control areas in HF [98] and have been shown to co-localize with activated neurons in the PVN, RVLM, and NTS of post-MI rats [78]. There is evidence to suggest that this transition takes place in response to augmented astrocytic and neuronal activation [98]. Summing up, chronic HF appears to trigger both regional (i.e., autonomic control areas) and global (i.e., cortex, hippocampus) chronic phenotypical and functional inflammatory changes in resident cells within the CNS.

As discussed above, neuroinflammation in HF may also result from peripherally activated immune cells entering the CNS [92]. CD4^+^ T-lymphocytes are globally expanded and activated in chronic ischemic HF [99]. Furthermore, the expression of tumor necrosis factor-α and tumor necrosis factor receptor increased in HF-activated CD4^+^ T cells [100]. Activated T cells were not only shown to cross the blood–brain barrier, but also to induce its disruption and increase its permeability [101]. In summary, processes from within the CNS (in response to increased neuronal activation) and from the periphery (i.e., activated T cells) seem to be intricately involved in heart failure-related neuroinflammation.

### 5.3. Hypertension

Models of hypertension increased the number of activated microglia in the cortex and hippocampus of mice, upregulated triggering receptor expressed on myeloid cells 2 (TREM2) produced by microglia, and increased amyloid-beta deposition [102]. In mice with transverse aortic constriction, it was demonstrated that hypertension per se triggered neuroinflammation before amyloid-beta deposition occurred [103]. Damage to the cerebral microvasculature and the locally activated renin–angiotensin system might thus be a crucial mechanism linking hypertension to neuroinflammation and neurodegeneration [104]. In turn, overstimulation of the pro-inflammatory pathways within brain areas responsible for sympathetic outflow is widely acknowledged as a primary contributing factor to the establishment and maintenance of neurogenic hypertension [105]. In another study, aerobic training regulated microglia activation and the production of pro-inflammatory cytokines in the presence of hypertension [106].

### 5.4. Atrial Fibrillation

Patients with AF exhibited elevated plasma concentrations of several inflammatory markers such as interleukin-1β and tumor necrosis factor-α that have been related to the development of AD [107]. Besides the generation of microemboli occurring with AF, an alternative explanation for the development of dementia in AF may be thrombotic microinfarctions triggered by inflammatory processes [108].

### 5.5. Aortic Valve Stenosis

Little is known about neuroinflammation in patients with aortic valve stenosis or its effects on microglia and astrocytes. Here, (pre-)clinical studies are needed.

### 5.6. Cardiac Arrest

Extensive microglial activation and neurodegeneration in the *cornu ammonis* area 1 (CA1) and the dentate gyrus of the hippocampus are evident following brief asystolic cardiac arrest and are associated with severe neurological injury [109]. Another recent study in rats demonstrated that microglial pyroptosis mediated by the NLR family pyrin domain containing 3 (NLRP3) inflammasome appears to be critically involved in the pathogenesis of post-cardiac arrest brain injury [110].

## 6. Potential Biomarkers

As shown above and summarized in Table 1, the evidence for neuroinflammation in cardiac diseases remains predominantly rooted on complex experimental findings. Considerable inter-individual differences, but also elaborate experimental settings and/or cost-intensive imaging (i.e., positron emission imaging), constitute current barriers for the advancement of this important research area. Therefore, biomarkers of cognitive decline and neuroinflammation might help in identifying patients at risk: studies of cardiac biomarkers to date have mainly focused on the N-terminal pro-B-type natriuretic peptide (NT-proBNP), which is an established surrogate of the severity of cardiac wall stress and symptom burden in HF. Of note, both cross-sectionally measured NT-proBNP and changes in NT-proBNP over time were found to predict incident dementia [111]. Furthermore, higher NT-proBNP levels were inversely associated with MMSE scores [112]. Interestingly, the measurement of highly sensitive troponins should also be considered an early and sensitive biomarker of cytotoxic effects of ‘inflammageing’ mechanisms (a term used to describe the chronic inflammatory state typical of elderly individuals) on myocardial tissue, as well as the cognitive decline in older adults [113].

Neuroinflammatory serum and cerebrospinal fluid (CSF) biomarkers associated with either microglia (soluble TREM2, monocyte chemoattractant protein-1 (MCP-1), and chitinase-3-like protein 1 (CHI3L1)) or astroglia (e.g., CHI3L1) are currently extensively investigated in AD patients [114]. Hence, these markers may reflect the inflammatory mechanisms within the CNS coupled with the neuro-degenerative pathways. A recent meta-analysis reported higher concentrations of CHI3L1, TREM2, MCP-1, and transforming growth factor-β in the CSF of AD patients compared to controls [115]. Along these lines, elevated neuronal serum biomarkers such as neurofilament light chain, phosphorylated tau protein and GFAP are associated with cognitive decline in chronic HF [116,117]. The recently observed increase in serum GFAP after MI might reflect the abovedescribed astroglial activation [118]. However, more studies on the mentioned biomarkers will be needed to further evaluate their relevance and potential in this currently evolving field of research.

## 7. Therapeutic Implications

From neurological research, there is now a group of candidate drugs that aim to modulate the activation and functional states of microglia, thus exerting potentially protective effects against the neuroinflammation caused by cardiac disorders.

Angiotensin II, a major hormone peptide that binds to angiotensin II type 1 and type 2 receptors (AT1R and AT2R, respectively) expressed in neurons, microglia, and astrocytes, has pleiotropic roles in the brain, including the mediation of inflammation and neuronal cell injury [119]. Interestingly, two antagonists of AT1R used to treat hypertension and heart failure, candesartan (NCT02646982) and telmisartan (NCT02085265), are now in phase II clinical trials for AD [120]. As these agents are already widely in use in cardiac disease, they might already make an important contribution to limiting cognitive functions.

Minocycline, an antibiotic that can permeate the blood–brain barrier, exhibits anti-inflammatory and neuroprotective effects [121]. Minocycline inhibits interleukin-1β, tumor necrosis factor, and interleukin-6 production by microglia in vitro, and suppresses microglial activation [122]. It also improves cognitive decline in AD transgenic mice [123]. However, preclinical or clinical data on the relevance of minocycline in cardiological diseases are missing.

In addition, pharmacological ubiquitin-specific protease 7 (USP7) inhibition attenuated microglia activation and associated neuron injury, thereby improving behavioral deficits in dementia and Parkinson’s disease mouse models. Thereby, the inhibition of USP7 might provide an attractive future direction for cardiogenic neurodegenerative disease [124].

## 8. Conclusions

There is now compelling and growing evidence for temporally and regionally varying, heterogenic neuroinflammatory changes occurring in the CNS during the progression of multiple cardiac diseases (Figure 1). These conditions are associated with accelerated cognitive decline beyond ageing and a higher prevalence of dementia. Recent findings from neurodegenerative research consistently suggest that chronic neuroinflammation precedes and aggravates cognitive decline, which might therefore also apply in cardiac disease. However, further (pre-)clinical investigations will be needed to specifically prove and evaluate the relative influence of neuroinflammation on cognitive decline in cardiological patients. So far, it has not been clear if and how reported regional changes in specific nuclei trigger global neuroinflammation, which is thought to precede cognitive decline. Furthermore, standardization of cognitive testing and better standardization of test settings (e.g., in-hospital vs. outpatient testing) will be important factors to advance research. Beyond contemporary guideline-based treatment of cardiac disorders, potential therapeutic options may include anti-inflammatory treatments specific to the brain in order to prevent an accelerated cognitive decline in cardiac disease.

## Figures and Tables

**Figure 1 life-13-00329-f001:**
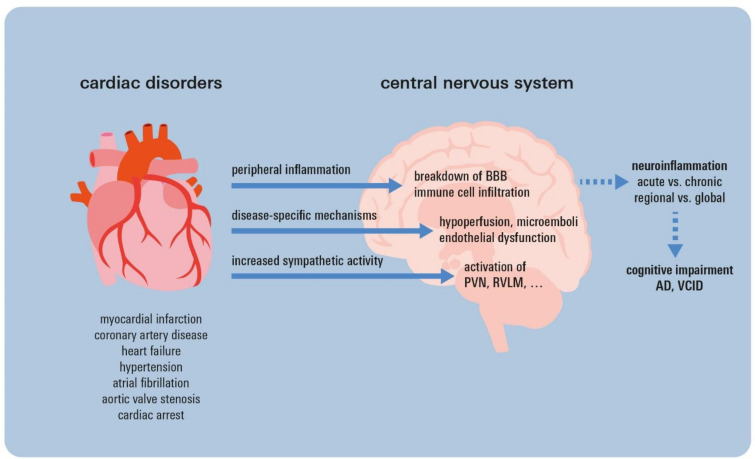
Current concept linking cardiac disorders, neuroinflammation, and cognitive impairment. For the displayed cardiac disorders, it has been shown that they can induce neuroinflammatory activation, thereby triggering cognitive impairment. Neuroinflammation may occur on both a regional and global level through (i) peripheral immune cell activation with subsequent CNS infiltration, (ii) disease-specific mechanisms, e.g., microemboli in atrial fibrillation, and (iii) over-activation of nuclei involved in sympathetic regulation. AD = Alzheimer’s dementia; BBB = blood–brain barrier; PVN = paraventricular nucleus; RVLM = rostral ventrolateral medulla; VCID = vascular cognitive impairment and dementia.

**Table 1 life-13-00329-t001:** Important preclinical and human studies on neuroinflammatory processes in cardiac diseases.

Cardiac Disease (Chapter in Article)	Evidence of (Chronic) Neuroinflammation		
	Preclinical Models	Human Studies	
Myocardial infarction/coronary artery disease (5.1)	Microglial activation in the RVLM, NTS, PAG, and PVN [77,78,79]	TSPO on microglia in the cerebellum, temporal and frontobasal cortex, and hypothalamus [75,76]	
	Phenotypic global changes to microglia [80,81,82]		
	Astroglial activation [76,84,85]		
	Monocyte and neutrophil abundance [87,88,89]		
Heart failure (5.2)	Inflammatory genes in the cortex and hippocampus [94]		
	Astrocyte activation in the RVLM [97]		
	Microglia co-localization and activation in the PVN, RVLM, and NTS [98]		
Hypertension (5.3)	Activation and TREM2 production by microglia in the cortex and hippocampus [102]		
	Neuroinflammation before amyloid-beta deposition [102,103]		
	Pro-inflammatory pathways involved in sympathetic control [105]		
Atrial fibrillation (5.4)	Interleukin-1β and tumor necrosis factor-α levels [107]		
Aortic valve stenosis (5.5)			
Cardiac arrest (5.6)	Microglial activation and neurodegeneration in the *cornu ammonis* area 1 [109]		
	NLRP3 inflammasome [110]		

## Data Availability

Not applicable.

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
