# Peer review of "Chronic Neuroinflammation and Cognitive Decline in Patients with Cardiac Disease: Evidence, Relevance, and Therapeutic Implications"

_life, 2023, doi:10.3390/life13020329_

Round 1
Reviewer 1 Report
The manuscript entitled, “Chronic neuroinflammation and cognitive decline in patients with cardiac disease: Evidence, relevance, and therapeutic implications" by Traub et al., aims to summarize the link between neuroinflammation and cognitive decline in patients with cardiac disease. Association between cognitive decline in patients with cardiac disease is well established, the factors inducing and accelerating cognitive decline beyond ageing and the intricate causal pathways and multilateral interdependencies involved is elaborated in this review article. Their study provides the current evidence linking cognitive impairment and chronic neuroinflammation in patients with various selected cardiac disorders including the aspect of chronic neuroinflammation as a potentially druggable target.
Overall, this is a well written, significant and well-timed article, this reviewer has certain recommendations that would assist to produce a more comprehensive overview of the topic:
Comments:
1 At least one illustrative figure may be provided as to highlight the summary of this study.
2, The English of manuscript can be polished (minor).
3, The authors should cross-check all abbreviations in the manuscript. Initially, define in full name followed by abbreviation.
4, Immune cells also play a crucial role in heart failure, specifically adaptive immune cells such as CD4 and CD8 etc. Authors should write a paragraph about immune cells and their effect on heart failure (PMID: 36093172; PMID: 36337927).
Author Response
The manuscript entitled, “Chronic neuroinflammation and cognitive decline in patients with cardiac disease: Evidence, relevance, and therapeutic implications" by Traub et al., aims to summarize the link between neuroinflammation and cognitive decline in patients with cardiac disease. Association between cognitive decline in patients with cardiac disease is well established, the factors inducing and accelerating cognitive decline beyond ageing and the intricate causal pathways and multilateral interdependencies involved is elaborated in this review article. Their study provides the current evidence linking cognitive impairment and chronic neuroinflammation in patients with various selected cardiac disorders including the aspect of chronic neuroinflammation as a potentially druggable target. Overall, this is a well written, significant and well-timed article, this reviewer has certain recommendations that would assist to produce a more comprehensive overview of the topic:
The authors thank for this positive comment.
Comments:
- At least one illustrative figure may be provided as to highlight the summary of this study.
We thank for this suggestion, which will help the reader understand the described interrelations more precisely. We added the following figure to our manuscript on page 8:
Figure 1: Current concept linking cardiac disorders, neuroinflammation, and cognitive impairment.
For the displayed cardiac disorders, it has been shown that they can induce neuroinflammatory activation thereby triggering cognitive impairment. Neuroinflammation my occur on both a regional and global level through i) peripheral immune cell activation with subsequent CNS infiltration, ii) disease-specific mechanisms, e.g. microemboli in atrial fibrillation, and iii) over-activation of nuclei involved in sympathetic regulation. AD = Alzheimer’s dementia; BBB = blood brain barrier; PVN = paraventricular nucleus; RVLM = rostral ventrolateral medulla; VCID = vascular cognitive impairment and dementia.
- The English of manuscript can be polished (minor).
In the revised version of the manuscript, we aimed to improve English throughout the text.
- The authors should cross-check all abbreviations in the manuscript. Initially, define in full name followed by abbreviation.
Thanks for this attentive remark. We cross-checked all abbreviations of the manuscript and explained them where they were first mentioned.
- Immune cells also play a crucial role in heart failure, specifically adaptive immune cells such as CD4 and CD8 etc. Authors should write a paragraph about immune cells and their effect on heart failure (PMID: 36093172; PMID: 36337927).
This is a very interesting topic. We added both references to the following new paragraphs on the role on T cells after MI and in chronic HF with regards to cognitive decline:
Lymphocytes are also likely involved in neuroinflammatory changes after MI. Recent studies suggested that CD4+ T lymphocytes exhibit biphasic kinetics post MI [90] with a rapid CD4+ T cell response at 3 days post-MI and a second phase of activation in chronic HF. Increased levels of pro-inflammatory markers after MI were shown to associate with worse systolic function [91]. It has also been established that peripherally activated T cells are capable of adhering to and crossing the blood–brain barrier to enter the perivascular space [92]. Murine and human immunohistochemical analysis showed a crucial role of perivascular and parenchymal infiltrating CD4+ helper T cells, but not cytotoxic T cells (CD8+) or B cells (CD20+), in the neocortex, hippocampus, and striatum in neurodegenerative disease [92], with close proximity to activated astroglia, microgliosis, and expression of pro-inflammatory cytokines. Therefore, peripherally activated T cells may also trigger or even amplify neuroinflammation after MI.
As discussed above, neuroinflammation in HF may also result from peripherally activated immune cells entering the CNS [91]. CD4+ T-lymphocytes are globally expanded and activated in chronic ischemic HF [98]. Further, the expression of tumor necrosis factor-α and tumor necrosis factor receptor increased in HF-activated CD4+ T cells [99]. Activated T cells not only were shown to cross the blood-brain barrier, but also to induce its disruption and increase its permeability [100]. In summary, processes from within the CNS (in response to increased neuronal activation) and from the periphery (i.e. activated T cells) seem to be intricately involved in heart failure related neuroinflammation.

Reviewer 2 Report
Great prospect for cardiac patients. The article is interesting and important.
Author Response
Great prospect for cardiac patients. The article is interesting and important.
Thanks for this positive feedback.

Reviewer 3 Report
The paper is focused on reviewing the current evidence linking cognitive impairment and chronic neuroinflammation in patients with various selected cardiac disorders including the aspect of chronic neuroinflammation as a potentially druggable target. It is a coherent topic to the aim and scope of the journal. The paper is well written and it follows the guidelines for authors of the journal.
I suggest to read and include the following paper Perrone MA, Aimo A, Bernardini S, Clerico A. Inflammageing and Cardiovascular System: Focus on Cardiokines and Cardiac-Specific Biomarkers. Int J Mol Sci. 2023 Jan 3;24(1):844.
Author Response
The paper is focused on reviewing the current evidence linking cognitive impairment and chronic neuroinflammation in patients with various selected cardiac disorders including the aspect of chronic neuroinflammation as a potentially druggable target. It is a coherent topic to the aim and scope of the journal. The paper is well written and it follows the guidelines for authors of the journal.
We appreciate this constructive and positive comment.
I suggest to read and include the following paper Perrone MA, Aimo A, Bernardini S, Clerico A. Inflammageing and Cardiovascular System: Focus on Cardiokines and Cardiac-Specific Biomarkers. Int J Mol Sci. 2023 Jan 3;24(1):844 36614282
Thanks for this thoughtful input. We have included this aspect (including reference) in our revised manuscript. The idea to discuss biomarkers motivated us to include a paragraph on biomarkers and cognitive decline in cardiac diseases on page 8:
Potential biomarkers. As shown above, the evidence for neuroinflammation in cardiac diseases remains predominantly rooted on complex experimental findings. Considerable interindividual differences, but also elaborative experimental settings and/or cost-intensive imaging (i.e. positron emission imaging) constitute current barriers for the advancement of this important research area. Therefore, biomarkers of cognitive decline and neuroinflammation might help identifying patients at risk: Studies of cardiac biomarkers until now mainly focused on the N-terminal pro-B-type natriuretic peptide (NT-proBNP), which is an established surrogate of the severity of cardiac wall stress and symptom burden in HF. Of note, both cross-sectionally measured NT-proBNP and changes in NT-proBNP over time were found to predict incident dementia [110]. Further, higher NT-proBNP levels were inversely associated with Mini-Mental State Examination scores [111]. Interestingly, the measurement of high-sensitive troponins should also be considered an early and sensitive biomarker of cytotoxic effects of ‘inflammageing’ mechanisms (a term used to describe the chronic inflammatory state typical of elderly individuals) on myocardial tissue, as well as the cognitive decline in older adults [112].
Neuroinflammatory serum and cerebrospinal fluid (CSF) biomarkers associated with either microglia (soluble TREM2, monocyte chemoattractant protein-1 (MCP-1), and chi-tinase-3-like protein 1 (CHI3L1)) or astroglia (e.g., CHI3L1) are currently extensively investigated in patients with AD patients [113]. Hence, these markers may reflect the inflammatory mechanisms within the CNS coupled with the neuro-degenerative pathways. A recent meta-analysis reported higher concentrations of CHI3L1, TREM2, MCP-1, and transforming growth factor-β in the CSF of AD patients compared to controls [114]. Along these lines, elevated neuronal serum biomarkers like neurofilament light chain, phosphorylated tau protein and GFAP associate with cognitive decline in chronic HF [115,116]. Recently observed increase of serum GFAP after MI might reflect above described astroglial activation [117]. However, more studies on the mentioned biomarkers will needed in to further evaluate their relevance and potential in this currently evolving field of research.
